# Heterogeneous Formation of DNA Double-Strand Breaks and Cell-Free DNA in Leukemia T-Cell Line and Human Peripheral Blood Mononuclear Cells in Response to Topoisomerase II Inhibitors

**DOI:** 10.3390/cancers16223798

**Published:** 2024-11-12

**Authors:** Christian Linke, Thilo von Hänisch, Julia Schröder, Werner Dammermann, Peter Markus Deckert, Mark Reinwald, Sandra Schwarzlose-Schwarck

**Affiliations:** 1Department of Hematology and Oncology, Center for Translational Medicine, University Hospital Brandenburg, Brandenburg Medical School Theodor Fontane, 14770 Brandenburg an der Havel, Germany; thilo.vonhaenisch@mhb-fontane.de (T.v.H.); julia.schroeder@mhb-fontane.de (J.S.); deckert@uk-brandenburg.de (P.M.D.); m.reinwald@klinikum-brandenburg.de (M.R.); s.schwarzlose-schwarck@uk-brandenburg.de (S.S.-S.); 2Department of Gastroenterology, Diabetology and Hepatology, University Hospital Brandenburg, Brandenburg Medical School Theodor Fontane, 14770 Brandenburg an der Havel, Germany; werner.dammermann@mhb-fontane.de; 3Faculty of Health Sciences, Joint Faculty of the Brandenburg University of Technology Cottbus-Senftenberg, the Brandenburg Medical School Theodor Fontane and the University of Potsdam, 14469 Potsdam, Germany

**Keywords:** DSB formation, γH2AX, cfDNA, topoisomerase II inhibitors, chemotherapeutic biomarker

## Abstract

Drug resistance and unwanted side effects in chemotherapies are major causes of therapeutic complications. To monitor cytotoxic drugs on a personalized scale, fast and easy applicable diagnostics should be developed. Chemotherapeutic agents often induce DNA damage, especially in rapidly dividing (cancer) cells, leading to its death. The detection of γH2AX, a protein involved in DNA damage repair, is considered as an efficient marker for drug-specific cell damage. Furthermore, cell-free DNA formation, a result of drug-induced cell death, may serve as a reliable diagnostic marker. Circulating in the bloodstream, cell-free DNA can be extracted in a minimally invasive way and repeatably quantified during chemotherapy. In time-course studies on drug-treated leukemia T-cells and healthy blood cells, both markers confirmed an enhanced drug response in dividing cells. Additionally, we showed formerly unknown drug-specific differences to timely induce DNA and cell damage. Our combined approach can improve personalized biomonitoring in chemotherapies and provides valuable information for patient biosampling.

## 1. Introduction

Chemotherapeutic drug resistance and a highly individual treatment response to antineoplastic agents, especially with regard to harmful side effects, pose great challenges for personalized cancer therapy. The detection of phosphorylated H2AX (γH2AX) is considered to be a sensitive biomarker of DNA damage, i.e., double-strand breaks (DSBs), in human cells and has been suggested for monitoring the sensitivity of malign and benign cells to cytotoxic agents, thus depicting an individualized patient response parameter [1,2,3]. Similar to γH2AX, cfDNA analysis emerges to be beneficial in the field of selection and adaptation of cancer therapy, treatment response and disease monitoring [4,5]. To our knowledge, a correlation between the two predictors has not yet been investigated.

Since the discovery that ionizing radiation can trigger a rapid phosphorylation of H2A histone variant H2AX at serine 138 by Ataxia-telangiectasia mutated (ATM) kinase, the phosphorylated form was named gamma-H2AX [6]. In the following years, γH2AX was shown to modulate both evolutionary conserved pathways of DSB repair, such as non-homologous end joining (NHJ) and homologues recombination (HR) [7,8,9,10]. A single DSB is believed to cause the phosphorylation of several hundreds of H2AX molecules, which can spread over approximately two Mbp of the adjacent chromatin region [6,11]. This accumulation can be detected with γH2AX-specific antibodies applying quantitative assays such as Western blotting, Fluorescence-activated Cell Sorting (FACS) or automated fluorescence microscopy [2,3,12,13,14]. To provide enough time for DNA repair, γH2AX mediates the recruitment of p53 binding protein 1 (53BP1) and the breast cancer gene protein 1 (BRCA1) to the DSB site, which, in turn, activate mechanisms to arrest the cell cycle [15]. A failure to repair DSBs induces the pro-apoptotic p53 upregulated modulator of apoptosis (PUMA) and apoptosis regulator Bcl-2-associated X protein (Bax). Especially in dividing cells, these events are of major importance for tumor suppression and are evolutionary highly conserved, which renders γH2AX as a critical mediator of genomic integrity, cell cycle checkpoints and cell death [16,17,18]. In addition to its function to serve as a platform for components of DNA damage response (DDR) and cell cycle checkpoint control, γ-phosphorylation of H2AX also plays an active role in apoptosis [6]. A careful examination by immunofluorescence microscopy revealed a staining pattern that differed completely compared to the focal-like distribution in DDR [19,20]. During early apoptotic events, γH2AX in nucleosomes localize to peripheral heterochromatic regions that are distributed at the inner nuclear membrane in a ring-like staining pattern [21]. These regions mark the early wave of DSBs in the sequence of internucleosomal DNA fragmentation mediated by apoptotic nucleases. Hereafter, the ring staining of γH2AX progresses to a diffuse overall staining of the entire nucleus, followed by the nuclear fragmentation and formation of apoptotic bodies [21].

In the last decade, several studies have examined the potential of γH2AX quantification to be applicable in therapeutic monitoring of DSB-inducing cytostatics such as etoposide, camptothecin, cisplatin, daunorubicin and cytarabine, among others [2,3,22]. However, only a few have simultaneously targeted the involvement of γH2AX in both DDR and apoptosis. Additionally, little is known about pharmacodynamics of antineoplastic drugs in actively replicating tumor cells versus terminally differentiated cells, which might be a prerequisite to assess drug efficacy on the one hand, and unwanted side effects on the other. To robustly quantify DDR and cell death in response to topoisomerase II inhibitors (etoposide, doxorubicin) on a single-cell basis, we applied automated microscopy (AKLIDES^®^ system, 2020AKL2-33). In addition, we utilized cell-free DNA (cfDNA) quantification via quantitative real-time PCR (qPCR) to investigate cellular degradation in response to the aforementioned drugs. This approach has been proven to biomark increased cell death events in the bloodstream of patients suffering from various diseases, including cancer [4,23,24,25]. Furthermore, it is assumed that this technique can differentiate between apoptotic and necrotic cell death if short- (~100 bp) and long-fragment (~250 bp) sizes of a specific cfDNA marker are amplified [26]. The ratio between long and short fragments represents the DNA integrity index (DII) and has been shown to differ between blood samples from cancer patients and healthy control cohorts in numerous studies [27,28,29]. In a common theory, it is assumed that fragments with a length of <180 bp are formed due to apoptotic nuclease activity [26]. In contrast, fragments > 250 bp are preferentially formed in tumor cells because of necrotic cell death caused by an active suppression of p53-mediated apoptosis [30]. Importantly, the immortalized T cell leukemia (Jurkat) cell line, which originally was isolated from the blood of a boy with acute lymphoblastic leukemia (ALL) in 1977, has been proven to harbor genetic mutations in TP53 and BAX, among others [31]. Accordingly, we also aimed to compare fragmentation of nuclear (Alu) and mitochondrial (MTCO3) cfDNA between proliferating Jurkat cells and PBMCs isolated from a healthy donor in response to cytostatics.

Our results showed significant kinetic differences in both DSB formation as well as cfDNA fragment quantities between both cell types, emphasizing the usefulness of our experimental approach in monitoring cytostatic effects more comprehensively. Our work also impacts the underrated issue of biosample collection timing, e.g. liquid biopsy, which, in a worst-case scenario, may lead to artificial results or unfavorable statistical power in clinical studies on cancer therapy.

## 2. Materials and Methods

### 2.1. Cells and Growth Conditions

The study was conducted in accordance with the Declaration of Helsinki, and local ethics committee approval was obtained. On the day of venipuncture, whole blood samples of a healthy donor were collected using CPT BD Vacutainer (Becton Dickinson, Franklin Lakes, NJ, USA). The blood samples were centrifuged at 1500× *g* for 20 min at room temperature (RT). PBMCs were isolated from buffy coat and washed twice with phosphate buffered saline (Gibco PBS pH 7.4 (1×), Fisher Scientific, Paisley, UK) by centrifugation at 350× *g* for 5 min at RT. The Jurkat cell line had been commercially obtained from Leibniz Institute DSMZ-German Collection of Microorganisms and Cell Cultures GmbH, Germany. Isolated PBMCs and Jurkat cells were seeded in T25 culturing flasks (Sarstedt, Nümbrecht, Germany) at a density of 3 × 10^6^ cells/mL using Dulbecco’s Modified Eagle Medium (DMEM GlutaMAX Gibco, Fisher Scientific, UK) supplemented with 10% fetal bovine serum (FBS) and antibiotics. Cells were cultured in a humidified atmosphere (37 °C and 5% CO_2_). Prior to application of ETP or DOX (both: Merck Healthcare GmbH, Taufkirchen, Germany), cells were incubated in fresh media for 30 min. ETP was dissolved in dimethyl sulfoxide (DMSO, Genaxxon Bioscience, Ulm, Germany), and DOX was dissolved in nuclease-free water (Qiagen, Hilden, Germany) to the indicated final concentrations before addition to the growth medium. Control cells were incubated in equivalent concentrations of DMSO. Cultures aliquots were sampled at indicated time points and stored on ice until further processing.

### 2.2. Automated Microscopy

Staining of PBMCs and Jurkat cells was performed using AKLIDES^®^ Nuk HLCC Kit (Medipan, Dahlewitz, Germany) with some modifications. Briefly, cells were washed with ice-cold PBS and fixed with 2% paraformaldehyde (PFA) at 4 °C for 30 min, washed twice with ice-cold PBS and permeabilized with 0.2% Triton X-100 for 5 min on ice. Two more washing steps with ice-cold PBS and subsequent blocking with 1% bovine serum albumin (BSA) in PBS followed. The cells were incubated with a commercially provided admixture of two primary antibodies, binding to γH2AX and 53BP1 (Medipan, Germany) in PBS with 1% BSA at the indicated dilutions for 1 h at room temperature. Two washing cycles with PBS were performed, followed by incubation with the respective FITC- and APC-conjugated secondary antibodies in the dark. After a final washing step with PBS, 25 µL of a cell suspension with 2 × 10^6^ cells/mL in PBS were transferred to a silanized glass slide, cells were allowed to settle for 15 min, PBS was then carefully removed with a paper towel and cells were covered with 4′,6-diamidino-2-phenylindole (DAPI)-containing mounting medium (Medipan, Germany).

For automated microscopy, we used the AKLIDES^®^ system (Medipan, Germany) as previously described in detail [3,32]. Briefly, based on an inverse fluorescence microscope (Olympus IX81, Olympus, Hamburg, Germany), this platform includes hardware and software to ensure fully automated imaging analysis and evaluation. Autofocusing based on DAPI staining identified cell nuclei using 10 images in the z dimension at a distance of 1 mm each throughout the nucleus. Predefined morphological parameters were a nuclear diameter between 4 and 50 µm and a nuclear convexity between 0.5 and 1.0. The focus diameter was defined between 0.1 and 1.2 µm. At least three technical replicates with a minimum of 60 cells each were stored as TIFF files of 60× magnification images and analyzed with the AKLIDES^®^ software. Exposure time for FITC- and APC-specific fluorescence varied between PBMCs (between 2000 ms and 800 ms) and Jurkat cells (between 1500 ms and 400 ms). Cells showing a pan-stained nucleus were scored as apoptotic cells when at least 70% of the DAPI stained area exceeded the foci threshold.

### 2.3. Fluorescence-Activated Cell Sorting (FACS)

To detect γH2AX by flow cytometry, we used the H2A.X Phosphorylation Assay Kit (Merck, Millipore, Burlington, MA, USA) according to manufacturer’s instructions. In brief, cultures aliquots obtained from time-course experiments were diluted to 1 × 10^6^ cells/mL in 350 µL PBS and treated with permeabilization solution (0.5% saponin, 10 mM HEPES, pH7.4, 140 mM NaCl and 0.25 mM CaCl_2_) at room temperature for 5 min, washed twice with PBS and stained with a FITC-conjugated anti-phospho-Histone H2A.X antibody. Cells were washed once with PBS and stained for DNA with Hoechst 33342 (Fisher Scientific, Eugene, OR, USA) in the dark at room temperature for 5 min. After final washing with PBS, cells were analyzed on a BD FACSCelesta™ (BD Biosciences, San Jose, CA, USA). For this, a compensation control was performed with either PBMCs or Jurkat cells treated with 100 µM ETP and stained (positive) or not stained (negative) for γH2AX. Additionally, all events were gated to cells positively stained for DNA applying BD FACSDiva™ software v9.0 (BD Biosciences, USA). The percentage of γH2AX positive events subtracted from overall events positive for Hoechst staining was used for statistical analysis.

### 2.4. Quantity and Integrity Index of Nuclear and Mitochondrial cfDNA

Medium supernatants of culture samples were centrifuged at 16,000× *g* for 10 min at 4°C, aliquots were transferred to 1.5 mL tubes and cfDNA concentration was determined using Qubit™ dsDNA HS Assay Kit and Qubit™ 3.0 Fluorometer (ThermoFisher, Fisher Scientific, Invitrogen, Eugene, OR, USA). Quantities of short and long fragments of nuclear (Alu) and mitochondrial (MTCO3) cfDNA were determined by quantitative real-time PCR (qPCR) on a Light-cycler 96 (Roche, Mannheim, Germany) using 2 µL of cell-free medium from the respective sample as a template. To minimize self-inhibiting of qPCR reactions by high sample concentrations, as had been observed at later time points (4–8 h), samples were diluted at least 1:2 with nuclease-free water with higher dilutions for later time points (4 h, 1:10; 6 h, 1:50; and 8 h, 1:100). A total of 15 µL reaction volume was employed in all qPCR analysis, containing 1× PowerUp™ SYBR™ Green Master Mix and 0.25 µM of primer. Cycling conditions consisted of initial denaturation at 95 °C for 2 min, and 40 cycles of of 95 °C for 15 s and 60 °C for 1 min. A standard curve with serial dilutions of genomic DNA was used to determine the concentration of nuclear and mitochondrial cfDNA as described previously [28]. The DNA integrity index was calculated as the ratio of long to short fragments (Alu 247/Alu 115 and MTCO3 296/MTCO3 67). Oligonucleotide sequences were described previously [28].

### 2.5. Statistical Analysis

For all statistical analyses we used GraphPad Prism (version 9.5.0, Graph Pad Software, Boston, MA, USA). Statistical significance of differences between pairs was assessed by Student’s *t*-test with a 95% confidence interval. Data are presented as the mean with standard deviation, and *p* values are displayed with asterisks (* *p* ≤ 0.05, ** *p* ≤ 0.01, *** *p* ≤ 0.001).

## 3. Results

### 3.1. Dose-Dependence of Drug-Induced γH2AX Formation in PBMCs

To first monitor dose-dependent γH2AX formation in response to cytotoxic drugs, PBMCs isolated from a healthy donor were cultured in medium with increasing concentrations of ETP (5, 10, 25, 50, 75 and 100 µM) and doxorubicin (0.05, 0.1, 0.5, 1.0, 5.0, 10.0) for 3 h. Since ETP was dissolved in DMSO, equimolar amounts of the dissolvent were used as a control. Cells were then fixated, stained for γH2AX and subjected to automated microscopy (AKLIDES system). The results, shown in Figure 1, demonstrate a dose-dependent increase in the proportion of fluorescence-positive cells with ETP-concentration, with a significant difference (*p* < 0.05) of the mean value over DMSO controls starting at 5 µM ETP. In response to DOX, the proportion of γH2AX-stained PBMCs notably rose above the control level at 0.05 µM, and a significant increase was found at a concentration of 1.0 µM DOX (*p* < 0.05), which also exhibited the peak in γH2AX formation. Analyzing samples of the same series by FACS, ETP treatment led to a continuous but insignificant increase in γH2AX-fluorescent cells from 25 µM up to 100 µM, whereas with DOX incubation γH2AX fluorescence seemed to decline compared to controls (Appendix A).

### 3.2. Kinetics of Drug-Induced γH2AX Formation in PBMCs and Jurkat Cells

In a previous study, ETP-treated PBMCs were reported to reach a maximum of γH2AX foci formation after 4–8 h [3]. Starting from this, we performed time-course analyses with Jurkat cells and freshly isolated PBMCs, incubated with either 25 µM ETP or 1.0 µM DOX and sampled after exposure times of 0, 0.5, 1, 2, 4, 6 and 8 h for γH2AX quantification against DMSO controls by both automated fluorescence microscopy (AKLIDES) and flow cytometry.

In automated microscopy, PBMCs treated with ETP displayed a visible increase in proportion of γH2AX-positive cells after 0.5 h of incubation, becoming significant after 1 h (*p* < 0.01) with a maximum after 2 h (Figure 2A). DOX treatment caused a visible and significant rise in positive cells only after 2 h (*p* < 0.05), with a (statistically not significant) peak after 4 h. With Jurkat cells, ETP reached a significant increase (*p* < 0.005) after 0.5 h with a maximum at 1 h, while with DOX, significant differences (*p* < 0.05) increased between 1 h and 2 h (Figure 2B). Starting at 4 h incubation time, fluorescence-positive cells also increased among controls, albeit without statistical significance, and in both PBMCs and Jurkat cells, the γH2AX signal in response to ETP and DOX seemed to decline. Quantification of γH2AX foci within cells revealed similar trends like the proportion of fluorescent-positive cells (Figure 2C,D). In DOX-treated PBMCs, foci increased after 2 h and peaked at 4 h without statistical significance. Similar to the number of fluorescent cells, a decrease in γH2AX foci numbers was observed at around 4 h for both ETP and DOX. Jurkat cells displayed a maximum point of foci numbers after 30 min (*p* < 0.05) of ETP treatment and after 2 h in response to DOX (*p* < 0.05).

Analyzing the same incubation series by FACS, disparate results were found for the two cell types: In PBMCs, ETP treatment led to a continuous but insignificant increase in γH2AX-fluorescent cells from 2 up to 8 h, whereas DOX incubation appeared to have no effect compared to controls (Appendix A). With Jurkat cells, however, we observed a significantly higher proportion (*p* < 0.01) of fluorescent cells after incubation with ETP, starting at 0.5 h (Appendix A). Other than in automated microscopy, neither PBMCs nor Jurkat cells displayed a decrease in γH2AX signal after longer incubation. Similar to the results obtained from FACS analysis on dose-dependent experiments, our data revealed no substantially enhanced fluorescence in cells treated with DOX. 

### 3.3. Kinetics of Drug-Induced 53BP1 in PBMCs and Jurkat Cells

In using the AKLIDES system, we measured 53BP1-specific fluorescence in PBMCs and Jurkat cells in analogous time-course experiments to γH2AX. In PBMCs, both the proportion of fluorescence-positive cells and the number of foci per cell gave a similar pattern: With ETP, a marked increase became visible after 0.5 h (significant only for foci), reaching its maximum (significant in both evaluations) at 1 h (Figure 3A,C). As with γH2AX, the logically expected constant plateau in fact leveled off after longer exposure times, albeit less so than with γH2AX. With DOX, increases in either evaluation of the 53BP1 signal were present after 2 h, without statistical significance compared to controls and without the tailing-off observed with ETP.

By contrast, Jurkat cells stressed with ETP revealed no significant differences in 53BP1 signal over controls measured either as the proportion of positive cells or as foci per cell (Figure 3B), although the foci per cell at 0.5 h displayed one of the highest mean differences, albeit with a broad error margin. DOX, however, measuring the proportion of positive cells, induced a maximum 53BP1 signal after 1 h, which—due to a lower background value—became significant after 2 h of incubation (*p* < 0.05) to taper off thereafter. This broadly matched the foci per cell, which also became markedly and significantly elevated over controls after 1 and 2 h.

We observed a considerably higher 53BP1 control background in Jurkat cells than in PBMCs when measuring the proportion of positive cells (Figure 3D). We also noticed a decline in foci number for both ETP and DOX treated cells after more than 1 h of incubation.

### 3.4. Kinetics of Drug-Induced Cell Death in PBMCs and Jurkat Cells

As γH2AX is actively involved in the induction of apoptosis [6], we utilized the capacity of the AKLIDES software to discriminate diffuse pan-nuclear γH2AX staining, indicating cell degradation, from the foci-like pattern originating from DDR.

In PBMCs, we noticed a marked but statistically insignificant increase in the proportion of apoptotic cells from 4 h to 8 h of incubation with ETP, but not with DOX (Figure 4A). 

With Jukat cells, an increase in diffuse apoptotic γH2AX became visible after 1 h of incubation with both ETP and DOX (Figure 4B). While not statistically significant in pairwise comparison, this result was found consistently for all later time points. The apoptotic response to DOX, however, was considerably weaker than to ETP. Jurkat cells displayed remarkably higher apoptotic γH2AX staining than PBMC.

We were also able to visually follow up the difference in γH2AX staining between PBMCs and Jurkat cells by assessing the pictures that were automatically collected by the AKLIDES system. Representative images are shown in Appendix A. Here, we noticed a co-localization of bright diffuse γH2AX staining and fragmentation of nuclei as indicated by DAPI staining.

### 3.5. Drug-Induced Kinetics of Nuclear and Mitochondrial cfDNA Formation

To compare the time series of cfDNA formation between PBMCs and Jurkat cells, fragments of nuclear (Alu 115 bp and Alu 247 bp) and mitochondrial marker (MTCO3 67 bp and MTCO3 296 bp) were quantified. Without drug treatment, PBMCs and Jurkat cells exhibited a continuous increase of n- and mtcfDNA levels, with Alu levels in Jurkat cells significantly higher at all time points compared to PBMCs (Appendix A). After 4 h incubation time, this difference became even more pronounced (*p* < 0.005). The Alu integrity index also increased gradually until 4 h in both cell types, with PBMCs displaying notably lower means without a statistically significant difference, while predominantly in Jurkat cells, a decrease was observed at later time points (Appendix A). No substantial difference in mt-cfDNA fragment levels and its corresponding DII was detected between PBMCs and Jurkat, but starting from 2 h of incubation, the DII tended to decrease in Jurkat cells, whereas a plateau was reached after 4 h in PBMCs (Appendix A).

Similar to the apoptotic γH2AX staining of ETP-stressed PBMCs, increased concentrations of Alu 115 and Alu 247 was found after 4 h of incubation when compared to the DMSO control (Figure 5A,B). By contrast, no difference was observed in response to DOX until 4 h, but then Alu levels decreased and became significant after 8 h of incubation compared to ETP (*p* < 0.05).

In drug-treated Jurkat cells, no substantial difference in the level of both Alu fragments was detected after 6 h, but then Alu fragments were observed at insignificantly higher levels compared to the control (Figure 5C,D).

A marked effect on mt-cfDNA formation was not observed in drug-exposed PBMCs (Figure 6A,B); nonetheless, starting at 1–2 h, DOX-treated Jurkat cells displayed considerable higher MTCO3 fragment levels (Figure 6C,D), with MTCO3 247 statistical significant in pairwise comparison to the control after 2 h (*p* < 0.05) and 6 h (*p* < 0.01).

### 3.6. Kinetics of Nuclear and Mitochondrial cfDNA Integrity Index

To investigate the kinetic impact of cytostatic drugs on cfDNA integrity index (DII) in PBMCs and Jurkat cells, the ratio between the long and small fragments of MTCO3 and Alu markers was calculated. While the DII in the control PBMCs rose from time point 0 h (0.86) to a plateau at 4 h (1.56), a lower mean was found after 1 h of incubation (0.81) in ETP-treated cells, with a gradual increase until 8 h (1.59) (Figure 7A). By contrast, incubation with DOX displayed an elevated DII already at time point 0 h (1.27) and was statistically significant after 30 min (*p* < 0.01) and 1 h (*p* < 0.005) compared to ETP-treated PBMCs. A constant increase was observed until 4 h (1.52), but the expected plateau leveled off after longer exposure times to a minimum (1.16) after 8 h of incubation. Similar to the DII of MTCO3, the DII of Alu in the control PBMCs rose over time; however, mean values were considerably lower, starting with 0.32 and reaching a maximum of 0.82 after 8 h (Figure 7B). While ETP showed no marked effect, the mean DII of DOX-treated samples steadily declined, starting from 0.57 at 1 h and reaching a minimum (0.35) after 8 h without statistical significance in pairwise comparison to the control.

With mean values close to 1.0, drug-treated Jurkat cells exhibited similar MTCO3 fragment ratios until 2 h; afterward, they reached a plateau (1.61) with ETP and 1.41 with DOX, albeit without statistical significance to the control levels (Figure 7C). Contrarily, analyzing the same time series in all samples, the Alu index tended to increase moderately until 4 h, with control cells starting at 0.72 and reaching a maximum of 1.1, but then tailing off to 0.73 (Figure 7D). Of note, the DII of ETP-treated samples also peaked at 4 h (1.3) and leveled notably, even though they were insignificantly higher after 6 h compared to the control, while with DOX, an increased DII (1.1) was found only after 6 h of incubation.

## 4. Discussion

Phosphorylation of H2AX is widely considered as a useful clinical biomarker for induction of DSBs caused by genotoxic stress such as exposure to radiation or cytotoxic drugs [21,33]. Recently, several assays have been developed to monitor γH2AX in tumor biopsies, circulating tumor cells and hair follicles of patients treated with topoisomerase I or PARP inhibitors [2,3,14,34,35]. It was proposed that γH2AX foci quantification may help to evaluate the efficacy of chemotherapeutic drug application, thus potentially contributing to a personalized drug developmental process [1,33]. 

Similar to γH2AX, the investigation of quantity and quality of cfDNA may provide information about a patients’ disease status, resulting in an increasing amount of literature aiming to improve molecular diagnostics based on liquid biopsy [4]. A large number of studies have focused on either γH2AX foci analysis or cfDNA quantification in PBMCs but have barely discussed potential dissimilarities between neoplastic and normal cells. Here, we mainly performed comparative kinetic analysis on formation of γH2AX and cfDNA in response to topoisomerase II inhibitors cultured with either the immortalized T-cell leukemia (Jurkat) cell line or healthy PBMCs. 

Our results with ETP and DOX point to a much faster DDR in Jurkat cells compared to PBMCs. Specifically, DSB formation after ETP treatment peaked significantly earlier in Jurkat cells (30 min) compared to PBMCs (2 h). Additionally, DOX-treated Jurkat cells exhibited increased γH2AX accumulation approximately 1 h earlier than in PBMCs, whereas the signal maximum was delayed around 2 h. We speculate that the different timing may partially be associated with an enhanced metabolic activity in Jurkat cells required for cell growth. In the presence of fetal calf serum (FCS)-supplemented mitogens, a cell division time of 20.7 +/− 2.2 h was reported for Jurkat E6.1 cell line [36]. Since T lymphocytes, the major cell fraction of PBMCs, were reported to be unaffected by FCS containing mitogens [37], our PBMCs were unstimulated and therefore largely quiescent. Thus, active DNA replication and transcription processes in dividing Jurkat cells should lead to enhanced DSB formation compared to non-dividing cells, as commonly suggested for application of topoisomerase II inhibitors in cancer therapy. Interestingly, we also detected a delayed formation of 53BP1 foci between both cell types, supporting our analysis on γH2AX. Similar to the γH2AX foci pattern, this gap appeared to be more pronounced in response to DOX, with maximum foci formation between 1 and 2 h for Jurkat cells and between 4 and 8 h in PBMC, while with ETP, a peak was detected after 30 min in Jurkat cells and after 1 h in PBMCs. At this point, we noted that DOX-treated Jurkat cells and ETP-treated PBMCs exhibited peak levels of 53BP1 foci approximately 0.5–1 h earlier compared to γH2AX specific signals. Since the recruitment of 53BP1 was reported to occur subsequent to the phosphorylation of H2AX, this unexpected result may have been due to increased ATM kinase activity posterior to a maximum binding of 53BP1 at DSB sites or a limited nuclear 53BP1 reserve. Note that cells were exposed to constant stress, and therefore a known competition between γH2AX and 53BP1 for the binding at tandem BRCT domain of mediator of DNA damage checkpoint protein 1 (MDC1) must be accounted for [38].

Compared to ETP, DOX application led to delayed DSB formation in both PBMCs and Jurkat cells. To the authors’ knowledge, a difference in the timing of cellular response between both topoisomerase II inhibitors has not been reported previously. For interpretation, we contemplate the individual pharmacodynamical properties: DOX belongs to the class of anthracyclines that intercalates with DNA, thus blocking its repair, replication and transcription [39]. By contrast, ETP inhibits the DNA unwinding function of topoisomerase II through direct protein interaction [40]. A divergent molecular action of both inhibitors to induce DDR was suggested by Huelsenbeck et al., who found a bell-shaped γH2AX foci formation in dose-dependent analysis on DOX-treated H9c2 cells, with a maximum at 1–2 µM [41]. Contrarily, ETP treatment led to a constant dose-dependent increase of H2AX phosphorylation, although a concentration of 100 µM was not exceeded [41]. Although we obtained similar results, we wondered whether our applied dose of DOX was less effective compared to ETP, thus leading to the observed timely delay in DDR. Following this argument, it seems unlikely to us because incubation with 1 µM DOX, but not with 25 µM ETP, caused peak phosphorylation of H2AX in dose dependent experiments. 

As aforementioned, little is known about the potential of apoptotic γH2AX signal formation as a biomarker for cell death, thus being useful to simultaneously assess DDR and cytotoxicity in studies on DSB-inducing drugs. Applying the AKLIDES platform and its integrated algorithms to differentiate between the foci-like and a diffuse overall nuclear staining of γH2AX in the course of cell death, we also found disparate results between PBMC and Jurkat cells. In ETP-treated PBMCs, cytotoxicity increased after 4–6 h and was more profound after 8 h of incubation, whereas with DOX, the level of pan-stained nuclei was comparable or even lower than in control cells. Importantly, our results obtained from the quantification of short and long fragments of Alu in PBMCs largely confirmed the imaging data, indicating higher ETP-specific cytotoxicity, albeit the above-mentioned DOX-specific delay in DDR probably led to a postponed cell death.

Applying automated microscopy, a considerable higher death rate was also observed in ETP-treated Jukat cells already after 2 h of incubation, whereas with DOX, the cellular death rate increased moderately. At this point, the quantification of Alu fragments did not support our imaging data because notably higher levels of Alu 115 and 247 were only found after 8 h. Nonetheless, compared to the control, we detected much higher levels of both mt-cfDNA fragments in DOX-treated Jurkat cells already after 1–2 h of incubation, indicating mitochondrial degradation. Of note, this result was more in line with our imaging data and points to enhanced cell death, probably due to DOX-specific intercalation with mtDNA. DOX was reported to generate reactive oxygen species, lipid peroxidation, membrane damage and mitochondrial degeneration [39,42,43]. In contrast, increased mt-cfDNA formation in DOX-treated PBMCs was not detected. Here, we believe that to sustain cell viability in dormant PBMCs, mitochondrial biogenesis, including mtDNA replication events, are not as important as in actively dividing Jukat cells. Interestingly, however, we found a higher ratio of long to short MTCO3 fragments (DII) at earlier time points, whereas the DII of control PBMCs and with ETP tend to increase over time. Since, upon DOX application, neither increased mtDNA fragment degradation nor cell death was found at later time points, we speculate whether this result may reflect a physiological protective drug response. In line with this argument, starting after 2 h of incubation, we also observed a remarkable decrease of the DII of Alu fragments until the end, potentially indicating reduced necrotic cell death events.

In drug-treated Jurkat cells, both the DII of MTCO3 and, to a minor extent, of Alu, were increased at later time points (4–8 h) when compared to the corresponding control, presumably indicating higher necrotic cell death events. Interestingly, we also generally observed a higher DII of Alu fragments (approx. 2-fold) in Jurkat cells compared to PBMCs with the exception of time point 8 h. These observations potentially reflect impaired apoptosis in the immortalized cell line due to genetic mutations in TP53 among others, as suggested previously [31]. Unexpectedly, we observed a reduced DII of Alu in control and drug-treated Jurkat cells after 6–8 h of incubation. Considering that we started our experiments with high cell concentrations (3 × 10^6^/mL), the nutrient supplements of growth medium might have been exhausted, or the pH may have turned out to be suboptimal, leading to lower cell division and therefore death rates. In fact, we observed a color change of phenol red in the growth medium, especially in DMSO-treated Jurkat cells after 8 h of incubation, indicating suboptimal culture conditions. Thus, we believed that Jurkat cell death analysis by quantification of Alu was not as effective as the automated imaging of γH2AX. Of note, Jurkat cells principally exhibited higher levels of n-cfDNA (>10 fold) than PBMCs, which may have reduced the accuracy of the qPCR results at later time points. In fact, to reduce artificial qPCR results, we had to dilute the template medium at later time points (4–8 h) to ensure equimolar concentrations in qPCR reactions, as stated in the method section. In addition to active cell division, Jurkat cells were reported to harbor a tetraploidic genome [31], which, after its degradation, may have also facilitated a higher n-cfDNA formation. 

Taken together, drug treatment of Jurkat cells initiates DDR earlier compared to PBMCs, with ETP inducing DBSs faster than DOX application. In addition, both automated microscopy as well as cfDNA analysis revealed an earlier drug-specific toxicity in Jurkat cells compared to PBMCs. Since DOX-treated PBMCs displayed no elevated cell death and a markedly decreased n-cfDNA integrity index at later time points, we speculate whether a protective cellular response to DOX potentially reduced cytotoxity until 8 h. In drug-treated Jurkat cells, higher cell death-specific γH2AX staining, increased mt-cfDNA fragment levels and a corresponding DII, suggest a higher efficacy of DOX application.

## 5. Conclusions

In conclusion, our results indicate that γH2AX and cfDNA are potent biomarkers to monitor individual therapeutic response prior and during chemotherapy, as for each, this has been independently suggested in previous studies [14,15,17,27,29]. Nonetheless, a combined monitoring of both markers in future clinical studies with patient material should prove a superior diagnostic outcome.

Since DOX application in actively dividing Jurkat cells significantly increased mt-cfDNA formation, we propose a robust chemotherapeutic effect on T-cell leukemia. With regard to unwanted side effects, however, an increased cardiotoxicity, partially due to an increased mitochondrial degeneration, must be anticipated [43]. In this regard, our kinetic data also provide evidence that mt-cfDNA quantification can be an effective approach to biomonitor modulators of DOX related cardiotoxicity.

Our results further demonstrate that the timing of a liquid biopsy in patient’s chemotherapy is of major importance for optimal biomarker diagnostics. For example, blood sampling associated with cell death analysis should be performed later compared to biomonitoring of DSB. Here, our work also aids in identifying suitable kinetics for clinical biosample acquisition to simultaneously screen the effects of DSB-inducing agents on DNA integrity and cell death more robustly. Aiming for a personalized cancer therapy based on cellular characteristics rather than standard criteria such as the body mass index, our work may also help to reduce sample size in clinical cohort studies and improve statistical power.

Finally, however, the limitations of the present study must be mentioned: We believe that, in our time-course experiments, at later time points, a lower Jurkat cell concentration would have been more favorable for robust qPCR analysis on n-cfDNA. Furthermore, we anticipated a delayed cytotoxicity of PBMCs in response to DOX beyond 8 h of incubation, which was not covered by our experimental design.

In summary, a co-investigation of γH2AX and cfDNA formation may improve personalized diagnostics in cancer therapy ranging from hematologic malignancies to solid tumors in a future perspective.

## Figures and Tables

**Figure 1 cancers-16-03798-f001:**
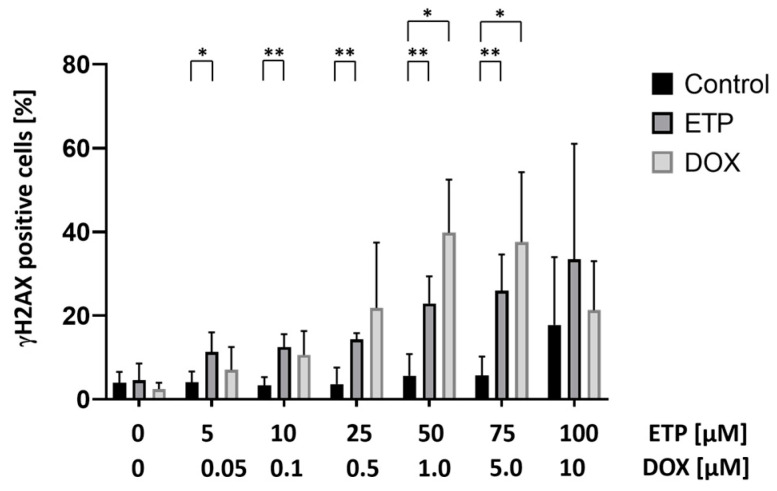
Dose-dependent γH2AX formation in PBMCs treated with increasing concentrations of ETP and DOX for 3 h: Immunofluorescent staining of phosphorylated H2AX was performed, and the percentage of γH2AX focus-positive cells was quantified by automated microscopy. PBMC treated with DMSO equimolar to the corresponding ETP concentration were used as controls. Data represent the mean and standard deviation of four independent experiments. Asterisks indicate significant difference between pairs as assessed by Students *t*-test: * *p* ≤ 0.05, ** *p* ≤ 0.01. ETP: etoposide; DOX: doxorubicin.

**Figure 2 cancers-16-03798-f002:**
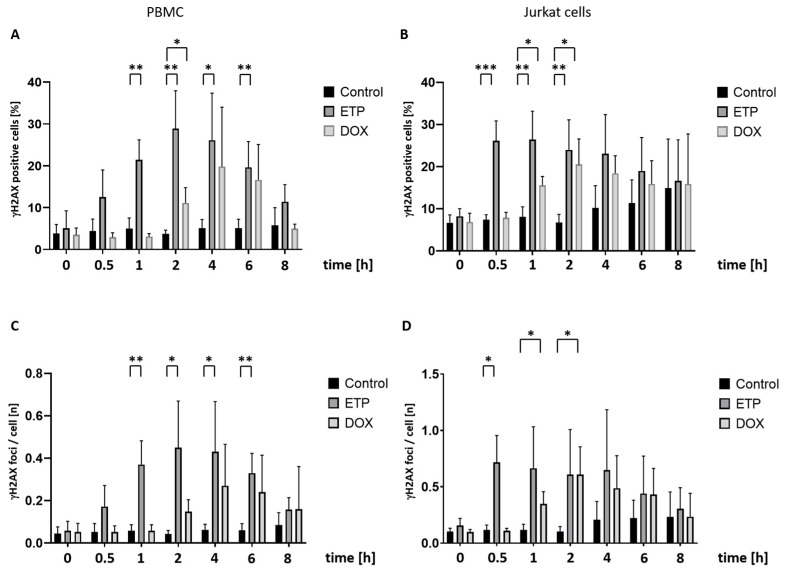
Time-dependent studies on γH2AX-specific staining in PBMCs and Jurkat cells after constant treatment with either ETP (25 µM) or DOX (1.0 µM): Automated quantification of cellular immunofluorescence between 0 and 8 h of incubation with topoisomerase II inhibitors using the AKLIDES system. DMSO-treated cells were used as a control. Percentage of γH2AX focus-positive cells in (**A**) PBMCs and (**B**) Jurkat cells were determined at the indicated time points; quantification of γH2AX foci number per cell in (**C**) PBMCs and (**D**) Jukat cells. Bars represent the mean and standard deviation of four independent experiments. Numbers of asterisks indicate increasing significance levels: * *p* ≤ 0.05, ** *p* ≤ 0.01, *** *p* ≤ 0.001. ETP: etoposide; DOX: doxorubicin.

**Figure 3 cancers-16-03798-f003:**
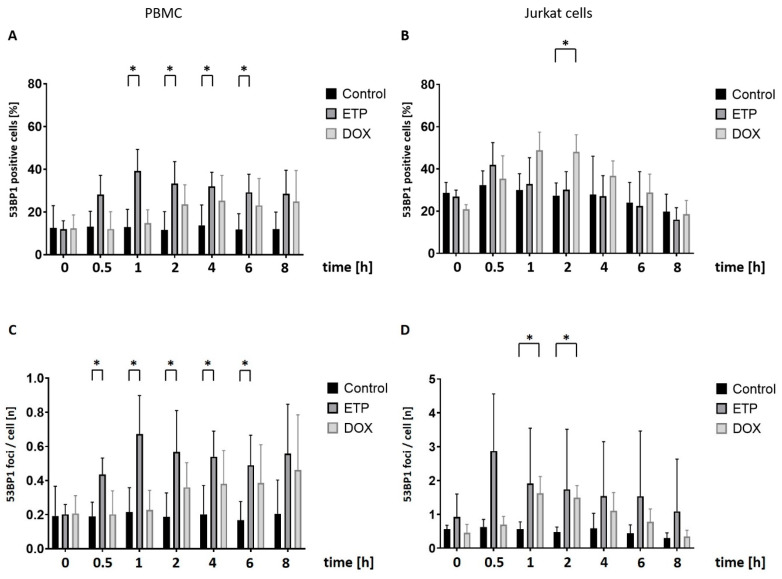
Time-dependent studies on 53BP1-specific staining in PBMCs and Jurkat cells after treatment with either ETP (25 µM) or DOX (1.0 µM): Automated quantification of cellular immunofluorescence between 0 and 8 h of incubation with topoisomerase II inhibitors using the AKLIDES^®^ system. DMSO-treated cells were used as a control. Percentage of 53BP1 focus-positive cells in (**A**) PBMCs and (**B**) Jurkat cells were determined at the indicated time points; quantification of 53BP1 foci number per cell in (**C**) PBMCs and (**D**) Jukat cells. Bars represent the mean and standard deviation of four independent experiments. Numbers of asterisks indicate increasing significance levels: * *p* ≤ 0.05. ETP: etoposide; DOX: doxorubicin.

**Figure 4 cancers-16-03798-f004:**
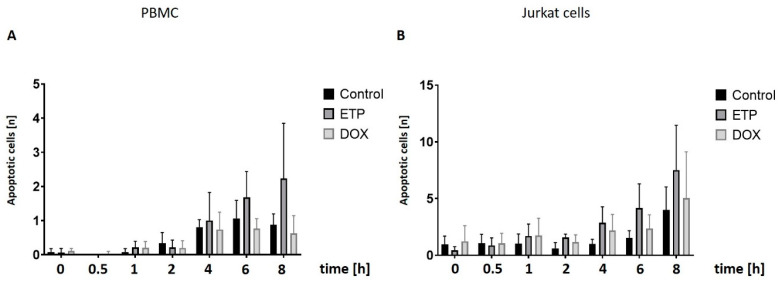
Time-dependent studies on apoptotic γH2AX staining in PBMCs and Jurkat cells treated with either ETP (25 µM) or DOX (1.0 µM): Nuclear pan-stained (apoptotic) γH2AX-specific immunofluorescence was analyzed by automated microscopy (AKLIDES). DMSO-treated cells were used as control. Numbers of apoptotic (**A**) PBMCs and (**B**) Jurkat cells were counted at the indicated time points. Data represent the mean and standard deviation of four independent experiments. ETP: etoposide; DOX: doxorubicin.

**Figure 5 cancers-16-03798-f005:**
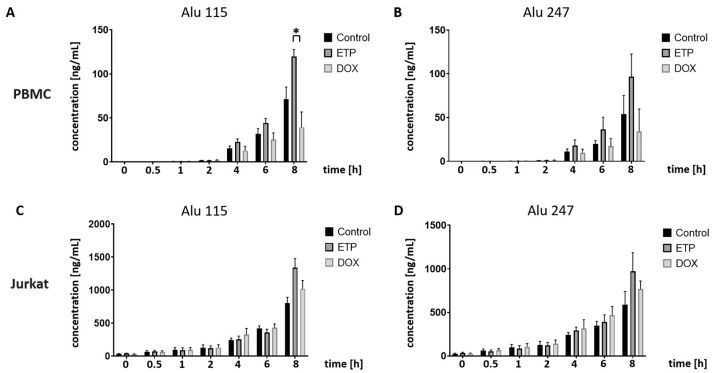
Time-dependent studies on nuclear cfDNA formation in PBMCs and Jurkat cells after treatment with ETP (25 µM) and DOX (1.0 µM): CfDNA fragment concentrations after qPCR analysis using supernatant medium at the indicated time points (0–8 h) as a template. DMSO-treated cells were used as the control. Nuclear cfDNA concentrations (in ng/mL) of (**A**) Alu 115 bp and (**B**) Alu 247 bp in PBMCs as well as (**C**) Alu 115 bp and (**D**) Alu 247 bp in Jurkat cells. Data represent the mean and standard deviation of four independent experiments. Numbers of asterisks indicate increasing significance levels: * *p* ≤ 0.05. ETP: etoposide; DOX: doxorubicin.

**Figure 6 cancers-16-03798-f006:**
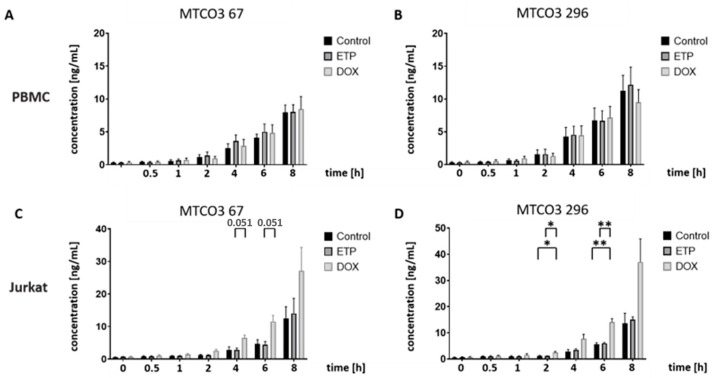
Time-dependent studies on mitochondrial cfDNA formation in PBMCs and Jurkat cells after treatment with ETP (25 µM) and DOX (1.0 µM): CfDNA fragment concentrations after qPCR analysis using supernatant medium at the indicated time points (0–8 h) as a template. DMSO-treated cells were used as the control. Mitochondrial cfDNA concentrations (in ng/mL) of (**A**) MTCO3 67 bp and (**B**) MTCO3 296 bp in PBMCs as well as (**C**) MTCO3 67 bp and (**D**) MTCO3 296 bp in Jurkat cells. Data represent the mean and standard deviation of four independent experiments. Numbers of asterisks indicate increasing significance levels: * *p* ≤ 0.05, ** *p* ≤ 0.01. ETP: etoposide; DOX: doxorubicin.

**Figure 7 cancers-16-03798-f007:**
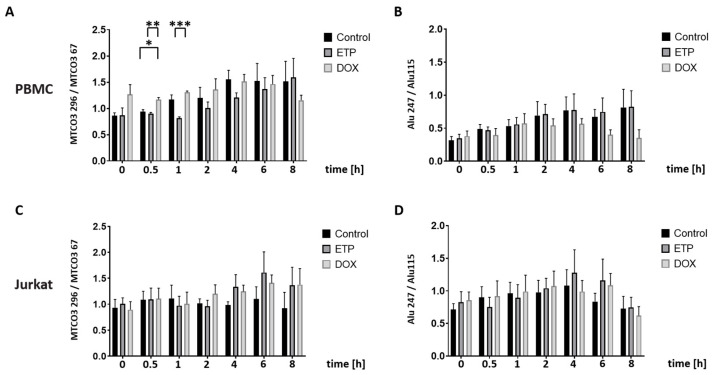
Integrity index obtained from nuclear and mitochondrial cfDNA fragment quantification in PBMCs and Jurkat cells after treatment with ETP (25 µM) and DOX (1.0 µM): DII values (dimensionless) of fragment concentrations from (**A**) MTCO3 and (**B**) Alu in PBMCs as well as (**C**) MTCO3 and (**D**) Alu in Jurkat cells. DMSO-treated cells were used as the control. Data represent the mean and standard deviation of four independent experiments. Numbers of asterisks indicate increasing significance levels: * *p* ≤ 0.05, ** *p* ≤ 0.01, *** *p* ≤ 0.001. ETP: etoposide; DOX: doxorubicin.

## Data Availability

The data presented in this study are available on request from the corresponding author due to ethical restrictions.

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
