# Peer review of "Heterogeneous Formation of DNA Double-Strand Breaks and Cell-Free DNA in Leukemia T-Cell Line and Human Peripheral Blood Mononuclear Cells in Response to Topoisomerase II Inhibitors"

_cancers, 2024, doi:10.3390/cancers16223798_

Round 1
Reviewer 1 Report
Comments and Suggestions for Authors
In the article " Heterogeneous formation of DNA double-strand breaks and cell-free DNA in leukemia T-cell line and Human Peripheral Blood Mononuclear Cells in response to Topoisomerase II inhibitors" C.Linke and co-authors analyze a number of leukemia-related factors that are of prognostic importance for evaluating the effectiveness of therapy with topoisomerase II inhibitors, such as etoposide and doxorubicin. These factors under study include the number of yH2AX–positive cells as an indicator of DNA double strand breaks, an assessment of the nuclear accumulation of 53BP1 protein as an important component of DSB repair, as well as a qPCR-based quantitation of cell-free DNA fragments. In the manuscript, the authors present a significant number of new results, undoubtedly of high scientific magnitude.
Importantly, the entire bulk of experimental data was obtained using two cell lines, namely Jurkat as a culture model of T-cell leukemia and PBMSs as healthy mononuclear cells. Such a comparative approach allows, on the one hand, to investigate the therapeutic effects of etoposide and doxorubicin as an anticancer agents, and on the other hand, to assess the degree of their side effects on untransformed cells of a similar nature.
Definitely, this article deserves to be published in in the journal Cancers. I have two minor technical comments on the text of the manuscript.
#1. Lines 62-64. “This accumulation can be detected with Æ´H2AX-specific antibodies applying quantitative immunohistochemical assays such as Western blotting, Fluorescence-activated Cell Sorting (FACS) or automated fluorescence microscopy”. I think, terminologically Western blotting cannot be classified as an immunohistochemical assay, because it deals with lysates, not cells. I would suggest to remove the word “immunohistochemical” from this sentence.
#2. The authors actively use numerous abbreviations, therefore I would suggest including a full list of abbreviations in the article for ease of reading.
Author Response
Comment 1: Lines 62-64. “This accumulation can be detected with Æ´H2AX-specific antibodies applying quantitative immunohistochemical assays such as Western blotting, Fluorescence-activated Cell Sorting (FACS) or automated fluorescence microscopy”. I think, terminologically Western blotting cannot be classified as an immunohistochemical assay, because it deals with lysates, not cells. I would suggest to remove the word “immunohistochemical” from this sentence.
Response 1: We thank the reviewer for pointing towards a wrong terminology, used in our manuscript and fully agree with your comment. We followed the suggestion and removed “immunohistochemical” from the respective sentence (page 2, line 63).
Comment2: The authors actively use numerous abbreviations, therefore I would suggest including a full list of abbreviations in the article for ease of reading.
Response2: The reviewer made an important point and we are glad to make the article more readable. Please find a full list of abbreviations in the appendix (page 15, line 601).
Reviewer 2 Report
Comments and Suggestions for Authors
Improving precision medicine in chemotherapy requires highly sensitive and easily applicable diagnostic tools. On the other hand, chemotherapeutic agents or cytotoxic drugs often induces DNA damage, especially in rapidly dividing cancer cells, as well as cell-free DNA formation, a result of drug-induced cell death. Both DNA damage and cell-free DNA formation may serve as reliable diagnostic biomarkers. In this manuscript, the authors employed automated microscopy and quantitative PCR-based analysis to investigate drug-induced DNA damage and cell-free DNA formation, and found previously unknown drug-specific differences to timely induce DNA and cell damage. Such combined approach may improve personalized biomonitoring in cancer chemotherapies and aid in decision for optimal patient biosampling. Overall, the manuscript is well prepared and the conclusions are generally supported by the information provided.
Comments:
1. Fig. 1 caption contains “*** P ≤ 0.001”, but the figure does not contain “***”; such statements should also be provided in other figure captions.
2. In Figures 2-6, they were time-dependent studies rather than real “kinetic studies” that can give kinetic constant for a process.
3. For practical applications, whether such combined detection of the both biomarkers could be efficient in blood samples?
4. There are some grammar and format errors, and some description is unclear. For example, “(c) Quantification of yH2AX foci number per cell in PBMCs and (d) Jukat cells” can be changed to “(c, d) Quantification of yH2AX foci number per cell in PBMCs (c) and Jukat (d) cells”.
Comments on the Quality of English LanguageThere are some grammar and format errors, and some description is unclear.
Author Response
Comment 1: Fig. 1 caption contains “*** P ≤ 0.001”, but the figure does not contain “***”; such statements should also be provided in other figure captions.
Response 1: Agreed. We are thankful for pointing to the unnecessary statement in Fig. 1 caption and missing information in subsequent figure captions. We fully corrected each of them in accordance with its content (Fig. 1, page 5, line 225; Fig. 2, page 6, line 269; Fig. 3, page 7, line 302; Fig. 5, page 9, line 360; Fig. 6 page 10, line 368; Fig. 7, page 10, 398; Suppl. Fig. 1, page 14, line 578; Suppl. Fig. 4, page 15, line 600).
Comment 2: In Figures 2-6, they were time-dependent studies rather than real “kinetic studies” that can give kinetic constant for a process.
Response 2: Thank you, we fully agree with your point. Therefore, we substituted the term “kinetic studies” with “time-dependent studies” or “time course studies” where it was applicable (page 4, line 166; page 5, line 229-230; Fig. 2, page 6, line 262; Fig. 3, page 7, line 295; Fig 4, page 8, line 321; Fig. 5, page 9, line 354; Fig. 6, page 9, line 362; page 13, line 537; Supl. Fig. 1, page 14, line 569; Suppl. Fig. 2, page 14, line 580; Suppl. Fig. 3, page 15, line 586).
Comment 3: For practical applications, whether such combined detection of the both biomarkers could be efficient in blood samples?
Response: 3 We are thankful to the reviewers comment, our preclinical ex-vivo (cell culture) study intend to analyse the efficacy of DSB-inducing agents in a cell model for T-cell leukemia and healthy PBMCs. The PBMCs used in our experiments were freshly prepared from blood samples of a healthy donor. However, drug treatment was performed posterior to the isolation procedure. Therefore, both approaches should be tested in a clinical setting using blood samples from (e.g. leukemia) patients treated with certain drugs. Accordingly, we stated: “Nonetheless, a combined monitoring of both markers in future clinical studies with patient material should prove a superior diagnostic outcome” (Conclusions, page 13, line 518-520).
Comment 4: There are some grammar and format errors, and some description is unclear. For example, “(c) Quantification of yH2AX foci number per cell in PBMCs and (d) Jukat cells” can be changed to “(c, d) Quantification of yH2AX foci number per cell in PBMCs (c) and Jukat (d) cells”.
Response 4: We are grateful to the reviewer for giving advice how to make our manuscript more reader-friendly and apologize grammar and formatting errors. Accordingly, we reviewed the whole manuscript carefully and corrected the indicated example as well as the following passages: Fig. 2, page 6, line 265-267; Fig. 3, page 7, line 298-300; Fig. 4, page 8, line 324; Fig. 5, page 9, line 357-358; Fig. 6, page 9, line 365-366; Fig. 7, page 10, line 395-396; Suppl. Fig. 1, page 14, line 371-373.
Reviewer 3 Report
Comments and Suggestions for Authors
In this study, the authors utilized automated microscopy to detect phosphorylated H2AX in response to drug treatment in both malignant T-cells and healthy cells. The results demonstrate that automated microscopy can accurately and promptly reflect DNA damage following drug treatment. Additionally, the authors examined the levels of nuclear and mitochondrial circulating free DNA (cfDNA) in both cell types after drug exposure. As expected, malignant T-cells exhibited a higher level of nuclear cfDNA compared to their healthy counterparts. Interestingly, treatment with DOX increased the level of mitochondrial cfDNA, indicating mitochondrial cfDNA may serve as a suitable biomarker for personalized medicine in response to DOX treatment.
Comments:
1. Did ETP and DOX induce H2AX phosphorylation directly or indirectly? What mechanisms are involved? Additionally, could wortmannin block H2AX phosphorylation in response to ETP and DOX treatment? Have the authors measured the total H2AX levels? It is presumed that the total levels should remain unchanged upon drug treatment.
2. The authors detected phosphorylated H2AX via immunofluorescence. Is it feasible to quantify fluorescence intensity using images captured by automated microscopy?
3. In Supplemental Figure 1, the authors measured phosphorylated H2AX using flow cytometry. Did the authors calculate the mean fluorescence intensity (MFI)?
4. In Figure 4, could the authors employ alternative methods to measure apoptosis, such as Annexin V and PI staining via flow cytometry, or detect cleaved Caspase-3 through immunoblotting? Additionally, could the authors clarify why the control (DMSO) appears to induce apoptosis in Jurkat cells when comparing 0 h to 8 h?
5. On line 345, Figures 6B and C should be corrected to Figures 6C and D. Furthermore, MTCO3 247 should be updated to MTCO3 296.
6. In Figure 1, the doses of 5, 10, and 25 µM of ETP show similar effects on the percentage of rH2AX-positive cells. Why did the authors choose the highest dose of 25 µM for subsequent studies? It might be more appropriate to select the lowest dose for clinical relevance.
Author Response
Comment 1: Did ETP and DOX induce H2AX phosphorylation directly or indirectly? What mechanisms are involved? Additionally, could wortmannin block H2AX phosphorylation in response to ETP and DOX treatment? Have the authors measured the total H2AX levels? It is presumed that the total levels should remain unchanged upon drug treatment.
Response 1: The reviewer made some important points and we are glad to answer his questions: In brief, etoposide inhibits the second step of Topoisomerase IIβ (TopIIβ) cleavage complex (i.e. DNA re-ligation). High-resolution of the ternary complex between TopoII, DNA and etoposide has revealed specific amino acids of TopoII that are critical for etoposide to interact with the TopoII-DNA complex. The drug by itself displays low affinity towards free DNA and is a poor DNA intercalator. A number of reports provide evidence that NHEJ is the predominant pathway for the repair of etoposide-induced TopoII-mediated DNA damage. This has been proven also in resting human T lymphocytes where etoposide treatment activates DDR and induces phosphorylation of ataxia telangiectasia mutated kinase (ATM) and of its substrates, H2AX and p53. The final effect is the activation of the pro-apoptotic PUMA (p53 upregulated modulator of apoptosis) protein and of caspases, leading to apoptotic cell death. However, recent findings suggest an involvement in autophagic pathways as well.
Doxorubicin intercalates with the DNA strands and inhibits TopIIβ: When doxorubicin intercalates with the DNA, complexes are formed leading to supercoiling of the DNA strands, which results in increased stress and unwrapping of the DNA from nucleosomes. Intercalation of doxorubicin also prevents TopIIβ from DNA binding, thus inhibiting the final DNA re-ligation step. In addition, doxorubicin is oxidized to semiquinone, an unstable metabolite, which is converted back to doxorubicin in a process that releases reactive oxygen species (ROS). In turn, ROS can lead to lipid peroxidation and membrane damage, DNA damage, oxidative stress, and triggers apoptotic pathways of cell death. Doxorubicin also accumulates in mitochondria, due to mtDNA intercalation and interaction with cardiolipin. The latter disrupts the organization of mitochondrial cristae and inhibits a proper function of the electron transport chain.
Thus, it seems that neither etoposide nor DOX induce DSBs directly, however DOX may have a stronger long-term effect on phosphorylation of H2AX by actively generating ROS, which in turn can lead to DNA damage.
Wortmannin is a known inhibitor of the phosphatidylinositol 3-kinase (PI3K) family, which includes double-stranded DNA dependent protein kinase (DNA-PK) and ATM. Thus, it is quite likely that H2AX phosphorylation would be blocked by wortmannin.
In our experiments, we did not measure the total amount of H2AX. Nonetheless, we share the reviewer’s opinion that H2AX biosynthesis is less likely a major part of DDR. In normal human fibroblasts, on average, there is one H2AX molecule every five nucleosomes. Thus, it appears that the pool of H2AX might be sufficient to cover the whole genome without the necessity for upregulation of H2AX level. Furthermore, as one of the fastest cellular response mechanism, we believe that H2AX biosynthesis would slow down the response to DSB induction, potentially with fatal outcome for genome integrity.
Comment 2: The authors detected phosphorylated H2AX via immunofluorescence. Is it feasible to quantify fluorescence intensity using images captured by automated microscopy?
Response 2: We are thankful to the reviewer’s comment. Indeed, using automated microscopy, Willitzki et al. 2013 demonstrated a linear increase of fluorescence intensity with ETP doses. In our study, we focussed on quantification of yH2AX-positive cells and foci number per cell. We believe that in contrast to fluorescence intensity, both parameters comprise a reliable differentiation between yH2AX-specific fluorescence originating from either DDR or upon cell death.
Comment 3: In Supplemental Figure 1, the authors measured phosphorylated H2AX using flow cytometry. Did the authors calculate the mean fluorescence intensity (MFI)?
Response 3: Thank you for pointing towards the possibility to calculate the mean fluorescence intensity in FACS experiments. As stated in the caption of Suppl. Figure 1, we calculated the percentage of yH2AX foci-positive cells, with bars representing the mean and standard deviation...” Nonetheless, we fully agree that the fluorescence intensity can be plotted as well. Unfortunately, DOX-specific background fluorescence interfered with the FITC-conjugated antibody in our FACS analysis, thus, we only were able to measure ETP and DMSO treated cells. Therefore, we decided not to perform additional analysis on FACS data.
Comment 4: In Figure 4, could the authors employ alternative methods to measure apoptosis, such as Annexin V and PI staining via flow cytometry, or detect cleaved Caspase-3 through immunoblotting? Additionally, could the authors clarify why the control (DMSO) appears to induce apoptosis in Jurkat cells when comparing 0 h to 8 h?
Response 4: Here, the reviewer made two important suggestions and we are grateful to the reviewer for giving advice how to support our findings with alternative methods. AnnexinV/PI staining and other methods were planned in the beginning but we decided not to continue with other approaches because of technical reasons: 1. The AnnexinV/PI protocol is based on incubation of (living) cells with the aforementioned fluorescent dyes at room temperature. However, our samples were kept on ice prior to fixation and yH2AX staining. To bring them back to room temperature followed by an incubation with AnnexinV/PI, the resulting time course experiments would not have shown the exact cell death rates at the indicated time points, as it was with yH2AX and cfDNA quantification. 2. Additionally, other approaches would have afforded to upscale the sample volumes (including the amount of cells and consumables), which unfortunately was an unfavourable cost factor to our budget. We deeply apologize for not having conducted additional experiments.
With regard to the second question, we believe that indeed the applied DMSO concentration (2.5 %) in the media effected the viability of Jurkat cells. Since it is known that DMSO increases cell permeability, most cell lines can tolerate only 0.5% DMSO and some cells can tolerate up to 1% without severe cytotoxicity.
Comment 5: On line 345, Figures 6B and C should be corrected to Figures 6C and D. Furthermore, MTCO3 247 should be updated to MTCO3 296.
Response 5: We are grateful to the reviewer for pointing to a wrong figure description and corrected the respective sentence (page 9, line 351).
Comment 6: In Figure 1, the doses of 5, 10, and 25 µM of ETP show similar effects on the percentage of rH2AX-positive cells. Why did the authors choose the highest dose of 25 µM for subsequent studies? It might be more appropriate to select the lowest dose for clinical relevance.
Response 6: Thank you, we fully agree with the reviewers comment. We applied 25 µM ETP for subsequent studies because it resulted best in FACS analysis on dose-dependent studies with PBMCs and Jurkat cells. Since the reviewer correctly pointed to this missing information, we decided to show the data in an updated version of Supplementary Fig. 1 (page 14, line 568).
Round 2
Reviewer 3 Report
Comments and Suggestions for Authors
Thank you to the authors for addressing my concerns.